# Effect of Intra- and Post-Operative Fluid and Blood Volume on Postoperative Pulmonary Edema in Patients with Intraoperative Massive Bleeding

**DOI:** 10.3390/jcm10184224

**Published:** 2021-09-17

**Authors:** Young-Suk Kwon, Haewon Kim, Hanna Lee, Jong-Ho Kim, Ji-Su Jang, Sung-Mi Hwang, Ji-Young Hong, Go-Eun Yang, Youngmi Kim, Jae-Jun Lee

**Affiliations:** 1Hallym University Medical Center, Department of Anesthesiology and Pain Medicine, Hallym University Chuncheon Sacred Heart Hospital, Chuncheon 24253, Korea; gettys@hallym.or.kr (Y.-S.K.); haewon@hallym.or.kr (H.K.); hanna@hallym.or.kr (H.L.); poik99@hallym.or.kr (J.-H.K.); jisu@hallym.or.kr (J.-S.J.); h70sm@hallym.or.kr (S.-M.H.); 2Institute of New Frontier Research Team, Hallym University, Chuncheon 24252, Korea; mdhong@hallym.or.kr (J.-Y.H.); Kym8389@hallym.ac.kr (Y.K.); 3Hallym University Medical Center, Division of Pulmonary, Allergy and Critical Care Medicine, Department of Internal Medicine, Hallym University Chuncheon Sacred Heart Hospital, Chuncheon 24253, Korea; 4Department of Radiology, Kangwon National University Hospital, Chuncheon 24289, Korea; yangke00@hanmail.net

**Keywords:** intraoperative massive bleeding, postoperative pulmonary edema, volume management, time varying hazard analysis

## Abstract

In patients with intraoperative massive bleeding, the effects of fluid and blood volume on postoperative pulmonary edema are uncertain. Patients with intraoperative massive bleeding who had undergone a non-cardiac surgery in five hospitals were enrolled in this study. We evaluated the association of postoperative pulmonary edema risk and intra- and post-operatively administered fluid and blood volumes in patients with intraoperative massive bleeding. In total, 2090 patients were included in the postoperative pulmonary edema analysis, and 300 patients developed pulmonary edema within 72 h of the surgery. The postoperative pulmonary edema with hypoxemia analysis included 1660 patients, and the condition occurred in 161 patients. An increase in the amount of red blood cells transfused per hour after surgery increased the risk of pulmonary edema (hazard ratio: 1.03; 95% confidence interval: 1.01–1.05; *p* = 0.013) and the risk of pulmonary edema with hypoxemia (hazard ratio: 1.04; 95% confidence interval: 1.01–1.07; *p* = 0.024). An increase in the red blood cells transfused per hour after surgery increased the risk of developing pulmonary edema. This increase can be considered as a risk factor for pulmonary edema.

## 1. Introduction

Massive bleeding during surgery can be fatal in the operating room and increases postoperative mortality [1]. Large volumes of fluid and blood, administered to compensate for the loss thereof, can result in fluid overload. Fluid overload, which is associated with increased hydrostatic pressure resulting in left ventricular dysfunction, is the most common cause of postoperative pulmonary edema [2]. Transfusions following massive bleeding can cause coagulopathies, acid–base abnormalities, hypothermia, and transfusion-related acute lung injury [3,4,5,6]. These complications of massive transfusions may also be related to pulmonary edema [7,8,9,10].

Intraoperative fluid therapy is complex and generally aims to meet maintenance requirements and to address existing fluid deficits and blood loss due to surgical wounds. Accurate evaluation of intraoperative blood loss volume [11] and safe transfusion can be challenging [12]. After surgery, further fluids and blood may be required due to postoperative bleeding and complications caused by massive intraoperative transfusions. Postoperative fluid administration and transfusions are even more difficult. Currently, there are no clinical approaches that ensure safety during fluid and blood administration, which are performed based on clinical approximations. Perioperative fluid management during massive intraoperative bleeding continues to be a challenge for surgeons and anesthesiologists.

In this study, we aimed to accurately determine the intra- and post-operative fluid and transfusion volumes. We also investigated the effects of intra- and post-operative transfusion volumes on the development of postoperative pulmonary edema in patients with massive intraoperative bleeding using a time-varying hazards analysis.

## 2. Materials and Methods

### 2.1. Data Collection

This retrospective cohort study was approved by the Clinical Research Ethics Committee of Chuncheon Sacred Heart Hospital, Hallym University. The study was conducted in accordance with the relevant guidelines and regulations of the committee. We enrolled at-risk individuals, such as brain trauma patients. The need for informed consent was waived because of the retrospective study design. The data were obtained from the clinical data warehouses of five hospitals at Hallym University Medical Center between 1 January 2010 and 31 December 2019. A clinical data warehouse is a database of medical records, prescriptions, and test results, which can be used to identify patients based on prescriptions, examinations, and diagnostic data. The timing and results of investigations, drugs administered, transfusion status, and other information can be extracted in an unstructured text format. The requested data are provided in a de-identified form, but the data for a specific person can be extracted using a key.

### 2.2. Patients

Bleeding was categorized into four classes based on the American College of Surgeons’ Advanced Trauma Life Support classification [13]. Class 4 corresponds to a loss of >40% of the circulating blood volume. In this study, an estimated blood loss >40% of the average blood volume (males: body weight (kg) × 75 mL; females: body weight (kg) × 65 mL) intraoperatively was defined as massive bleeding, and patients with massive intraoperative bleeding were included. The exclusion criteria were as follows:Age < 18 years;Patients undergoing cardiac surgery;Patients who underwent a previous surgery within 7 days of the current surgery; Patients with preoperative pulmonary edema or hypoxemia (PaO_2_/FiO_2_ ≤ 300).

Although arterial blood gas (ABG) data could not be obtained, patients who did not receive oxygen therapy postoperatively due to an absence of respiratory symptoms and those who did not have evidence of pulmonary edema on a chest X-ray were considered non-hypoxemic.

### 2.3. Primary and Secondary Outcomes

The primary outcome was the presence of pulmonary edema on a postoperative chest X-ray. Chest X-rays, evaluated by a radiologist, were used for consistency. The secondary outcome was postoperative pulmonary edema with hypoxemia. Hypoxemia was defined as PaO_2_/FiO_2_ ≤ 300. The results with the shortest interval between the chest X-ray and ABG analysis were used. Patients who had no pulmonary edema on the chest X-ray and who did not receive oxygen postoperatively were considered to be free of pulmonary edema. As pulmonary edema can occur up to 3 days postoperatively, it was evaluated until 72 h after surgery.

### 2.4. Major Variables

The major variables evaluated in this study were divided into intra- and post-operative variables. Intraoperative variables included the total and hourly amounts of fluid, red blood cells (RBCs), and fresh frozen plasma (FFP) administered during surgery.

Postoperative variables included the total and hourly amounts of fluid, RBCs, and FFP administered after surgery. All of the major postoperative variables were time-varying variables. The amounts of fluid and blood administered after surgery were measured based on the ABG analysis or chest X-ray times. For the analysis of pulmonary edema with hypoxemia, in which both an ABG and chest X-ray were performed, the ABG analysis time was used for the analysis, unless there were no respiratory symptoms and only the chest X-ray test was performed, in which case the chest X-ray time was used. The administered fluid and blood volumes are expressed as percentages of the patient’s average blood volume. Each patient had 1–3 time-varying variables and observation periods, depending upon the number of measurements (which varied according to the test frequency). The first observation period was between the end of anesthesia and the first test; the second was between the first and second tests; and the third was between the second and third tests.

### 2.5. Other Variables

Other covariates were adjusted to prevent residual confounding and biases. Demographic variables included old age (≥70 years), male sex, and obesity (body mass index ≥ 30). Preoperative variables included emergency, an American Society of Anesthesiologists physical status >2, smoking, brain trauma, multiple fractures, hyponatremia (<135 mmol/L), hypoalbuminemia (<3.5 g/dL), and glomerular filtration rate (GFR). Intraoperative variables included general anesthesia, anesthesia time, type of surgery (acute abdomen, aorta, brain, spine, thoracic), massive transfusion (packed RBCs ˃ 4 pints/h) [14], urine output < 0.5 mL/kg/h, continuous inotrope use, and estimated blood loss. Postoperative variables included patient-controlled analgesia and whether or not creatinine increased by 0.3 mg/dL compared to before surgery [15].

### 2.6. Statistical Analysis

Continuous variables are expressed as medians and interquartile ranges due to skewness. Categorical variables are expressed as frequencies and percentages. Continuous data were analyzed using the Mann–Whitney test to compare patients with and without postoperative pulmonary edema. Categorical data were analyzed using the chi-square test. Cox’s time-varying hazards model was used to obtain the hazard ratios of the major variables with respect to the occurrence of postoperative pulmonary edema within 72 h after surgery. Time-varying covariance occurs when a covariate changes over time during the period in which it is tracked. These variables can be analyzed with a time-varying Cox regression model to estimate their effect on event occurrence time [16]. A subgroup analysis of emergency surgery patients was also performed. Fluid and blood cut-off values for the occurrence of pulmonary edema were calculated using a receiver operating characteristic (ROC) curve analysis. All reported *p*-values were two-sided, and a *p*-value < 0.05 was considered to indicate statistical significance. SPSS software (version 24.0; IBM Corp., Armonk, NY, USA) was used for all statistical analyses except for time-varying Cox regression analysis, which was performed using Anaconda (Python version 3.7; Anaconda Inc., Austin, TX, USA) and Lifelines package (version 0.24.15; Accessed on 14 September 2021 https://github.com/CamDavidsonPilon/lifelines/blob/master/docs/index.rst).

## 3. Results

### 3.1. Study Population

Between 1 January 2010 and 31 December 2019, a total of 2161 patients had a bleeding amount >40% of their average blood volume during surgery and were included in the study. We excluded 71 patients because of missing data, and the remaining 2090 patients were included in the primary outcome analysis. There were one, two, and three follow-up periods for 638, 407, and 1045 patients, respectively. Furthermore, 300 patients had pulmonary edema findings on chest X-rays (Table 1). In the analysis of pulmonary edema with hypoxemia, 430 patients were excluded due to a lack of PO_2_ or FiO_2_ data, and the remaining 1630 patients were included. There were 161 patients with postoperative pulmonary edema with hypoxemia (Table 2). There were one, two, and three follow-up periods for 848, 499, and 313 patients, respectively.

### 3.2. Follow-Up Period

The total observation period was divided into six periods based on the start and end of the follow-up observation for the evaluation of the variables of interest (Table 1 and Table 2). Depending on the number of observations, patients could be included in several periods (Table 1 and Table 2). The median interval between the chest X-ray and ABG analysis was 2.4 h (interquartile range: 1.1–5.6 h).

### 3.3. Hazard Ratios for Postoperative Pulmonary Edema with and without Hypoxemia

The unadjusted and adjusted hazard ratios (aHRs) of major and kidney-related variables (preoperative GFR, intraoperative urine output ≤ 0.5 mL/kg/h, and postoperative creatinine increase) for 72-h postoperative pulmonary edema both with and without hypoxemia are shown in Figure 1 and Figure 2, respectively. The postoperatively administered total fluid (aHR: 1.00; 95% confidence interval CI: 1.00–1.00; *p* < 0.001) and the postoperative RBCs per hour (aHR: 1.03; 95% CI: 1.01–1.05; *p* = 0.013) showed significant associations with postoperative pulmonary edema. The postoperatively administered total fluid (aHR: 1.00; 95% CI: 1.00–1.00; *p* = 0.005) and the postoperative RBCs per hour (aHR: 1.04; 95% CI: 1.01–1.07; *p* = 0.024) also showed significant associations with postoperative pulmonary edema with hypoxemia. The intraoperative FFP per hour was significantly associated with postoperative pulmonary edema (aHR: 1.03; 95% CI: 1.00–1.06; *p* < 0.042), but the other major variables were not (Figure 1 and Figure 2; Appendix A, Table A1 and Table A2).

### 3.4. Blood and Fluid Cut-Off Values for Pulmonary Edema with and without Hypoxemia

The cut-off values of blood and fluid for the development of pulmonary edema with and without hypoxemia are summarized in Table 3.

### 3.5. Emergency Surgery

The aHRs of major and kidney-related variables for 72-h postoperative pulmonary edema with and without hypoxemia are shown in Table 4.

## 4. Discussion

Through time-varying Cox regression, it was shown that the total amount of fluid administered after surgery and the amount of RBCs administered per hour were associated with an increased postoperative risk for pulmonary edema, both with and without hypoxemia. However, the association with the amount of RBCs had low statistical significance for emergency surgeries. The hazard ratio of the total amount of fluid administered postoperatively for pulmonary edema was too small to be clinically significant.

Our findings showed that in patients with massive intraoperative bleeding, the amount of RBCs administered per hour postoperatively was significantly associated with the risk of postoperative pulmonary edema. Exacerbations of pre-existing anemia, intraoperative bleeding, and repeated laboratory blood tests were common causes of postoperative anemia [17,18,19]. More RBC transfusions may be required for anemia that is severe or difficult to correct. Anemia and pulmonary edema are interrelated, as low hemoglobin levels can cause salt and water retention, hormonal and metabolic changes, myocardial toxicity, and myocardial hypertrophy [20,21,22,23,24,25,26]. Heart failure can lead to anemia through various mechanisms, such as iron deficiency, inflammation, low erythropoietin levels, medications, hemodilution, and medullar dysfunction [27].

The principle of fluid balance is that the amount of fluid administered should be equal to the amount of fluid lost [28]. However, weight increase in surgical patients is common, typically due to a positive fluid balance [29,30]. Cooperman et al. reported 2.1 ± 0.4 L of bleeding and 3.7 ± 0.6 L of fluid replacement in postoperative pulmonary edema patients, with half of the patients exhibiting manifestations of fluid overload [31]. Arieff et al. reported an increased risk of postoperative pulmonary edema when fluid retention exceeding 67 mL/kg/day [32]. Holte et al. reported that the infusion of 40 mL/kg of lactated Ringer’s solution for 3 h reduced pulmonary function [33]. Massive bleeding also causes fluid imbalances, and massive volumes of fluid and blood are rapidly administered in such cases. Massive transfusions can cause circulatory overload [14], which can, in turn, cause postoperative pulmonary edema [2,31,32,34].

In this study, intraoperatively administered fluids and blood were not related to the risk of postoperative pulmonary edema with or without hypoxemia. Perioperative fluid and blood transfusions can cause hypothermia [35,36,37], which may lead to impaired coagulation and acidosis [38,39] as well as pulmonary edema [9]. Fluid and blood administration following bleeding can cause dilutional coagulopathies, platelet dysfunction, fibrinolysis, and hypofibrinogenemia [3,40,41]. These coagulation disorders are present in many patients after surgery and exacerbate bleeding [42,43]. The exacerbated bleeding and resultant anemia increase the need for postoperative blood transfusions [44]. This increases the risk of developing transfusion-related acute lung injury and circulatory overload [45,46]. Acidosis and hypocalcemia caused by blood transfusions can affect heart contractility and can cause pulmonary edema [47,48,49]. However, correctable abnormalities may not last long because the patient is closely monitored intraoperatively for abnormalities such as anemia [50] as well as for volume status [51], body temperature, electrolytes, and coagulation defects. Immediate correction is attempted in the event of abnormalities [35,36,37,38,39,52]. Pulmonary edema is not visible on chest X-rays until the amount of lung fluid increases by 30% [53] and may be missed during intraoperative monitoring unless there is a sudden increase in lung fluid. However, in patients with massive bleeding, if sufficient resuscitation is not achieved during surgery, organ failure may occur [54] and could lead to postoperative pulmonary edema [55,56,57]. Frequent postoperative volume monitoring using readily available clinical parameters, such as vital signs, urine output, lung auscultation, weight change, and net fluid retention [58,59], is therefore recommended. However, this is rarely undertaken in a clinical setting before pulmonary edema is diagnosed [32].

In this study, increased hourly FFP administration during surgery increased the risk of postoperative pulmonary edema. While FFP is used in cases of clotting factor deficiencies to prevent hemorrhage, it puts patients at risk for transfusion-related acute lung injury [6] and can cause circulatory overload in patients with kidney or cardiopulmonary failure [60]. However, the lack of association between hourly FFP administration during surgery and the risk of postoperative pulmonary edema with hypoxemia requires further investigation.

Kidney function is also closely related to the development of pulmonary edema [57,61,62]. We evaluated renal function in terms of the preoperative GFR, intraoperative oliguria, and postoperative creatinine and found that increased postoperative creatinine increased the risk of postoperative pulmonary edema. The kidneys are involved in various homeostatic mechanisms and interact with the lungs by regulating the acid–base balance, increasing oxygen-carrying capacity through RBC production, and regulating blood pressure through the renin–angiotensin–aldosterone axis. However, as acute kidney injury (AKI) progresses, these processes can become impaired [57]. Intraoperative anemia, shock, and hemodilution increase the risk of postoperative AKI [63,64] and are more likely to occur in patients with massive intraoperative bleeding [65,66,67]. Fluid retention may occur during anesthesia because the blood vessels expand [68], and the short operation times make it difficult to observe kidney damage due to oliguria during surgery [69]. We determined whether a creatinine increase of 0.3 mg/dL had occurred compared to the preoperative baseline, but this does not necessarily imply AKI. AKI is defined as an abrupt (within 48 h) reduction in kidney function, currently defined as an absolute increase in serum creatinine of ≥0.3 mg/dL (≥26.4 μmol/L), a percentage increase in serum creatinine of ≥50% (1.5-fold from baseline), or a reduction in urine output (documented oliguria < 0.5 mL/kg/h for >6 h). An increase in creatinine could not be defined as AKI because not all of the perioperative parameters that were measured were taken within 48 h. However, we tried to evaluate the possibility of kidney damage through the increase in creatinine.

In this study, it was observed that postoperative pulmonary edema may occur in a significant proportion of patients with massive intraoperative bleeding. This may occur in patients with or without serious cardiovascular disease, in which fluid overload in the postoperative period may be a significant factor. Alternatively, negative pressure pulmonary edema may develop, which may be a different and important clinical entity in the period immediately after extubation and may progress rapidly. Negative pressure pulmonary edema may result from acute upper airway obstruction following the creation of acute negative intrathoracic pressure. Although negative pressure pulmonary edema can be a life-threatening complication during surgery without timely diagnosis and treatment, it is recognized as having a good prognosis if recognized immediately and if treated appropriately [70]. In further studies, discussion on the existence of negative pressure pulmonary edema in the development of postoperative pulmonary edema is needed.

In this study, time-varying Cox regression was used to obtain hazard ratios. Time-varying covariance is commonly seen in clinical studies when a given covariate changes over time during the follow-up period [16,71]. In cases with massive intraoperative bleeding, the patient’s condition after surgery may change in various ways; based on these changes, several tests may be performed. The fluid and transfusion volumes are also modified accordingly. In such cases, a time-varying model may be appropriate. We calculated the cut-off values of blood and fluid for the development of pulmonary edema with and without hypoxemia, which showed significant differences. These differences may have occurred because pulmonary edema with hypoxemia is often more severe. However, the area under the ROC curve value was not excellent, and the individual cut-off values may be unreliable.

### Limitations

Although this study included a large number of patients with massive intraoperative bleeding, it is difficult to perform a randomized controlled trial of massive bleeding patients. This study used a retrospective design, which is subject to bias. Pulmonary edema is either cardiogenic or non-cardiogenic, but we could not stratify the patients based on the type of pulmonary edema. Although the different types of pulmonary edema have different causes, it is difficult to distinguish them because of their similar clinical features. In addition, the causes can be complex in critically ill patients [55], and both types of postoperative pulmonary edema can occur in patients with massive intraoperative bleeding [34,45,46]. The gold standard for determining the cause of acute pulmonary edema is the insertion of a pulmonary artery catheter, but this is not routinely performed [72]. Instead, in this study, we assessed pulmonary edema using less invasive methods. Chest X-rays have certain limitations for diagnosing pulmonary edema. Even if mild pulmonary edema is depicted on an X-ray, surgery is often performed if there are no symptoms. However, certain radiographic features may help determine the cause of the edema [55,73]. In addition, hypoxemia was also analyzed to evaluate the severity of pulmonary edema symptoms. Finally, factors related to pulmonary edema, such as urine output, intraoperative blood pressure, and base excess were insufficient. Except for some serious cases admitted to the intensive care unit, no data on postoperative fluid output were available for the follow-up period. Instead, we assessed kidney function based on changes in creatinine levels. Blood pressure data and base excess during surgery could not be included in the analysis due to difficulties in data extraction. However, the use of intraoperative inotropes was included and was evaluated as an indirect factor.

## 5. Conclusions

This study demonstrated that the incidence of postoperative pulmonary edema is higher in patients with massive intraoperative bleeding and in those receiving large RBC transfusions postoperatively. These findings are in agreement with previous studies. In patients with massive intraoperative bleeding, close monitoring and rapid correction of the causes of postoperative anemia and bleeding may be more important. Further prospective population-based studies are needed to explain the mechanisms underlying the association between postoperative RBC transfusion and the risk of developing postoperative pulmonary edema in patients with massive intraoperative bleeding.

## Figures and Tables

**Figure 1 jcm-10-04224-f001:**
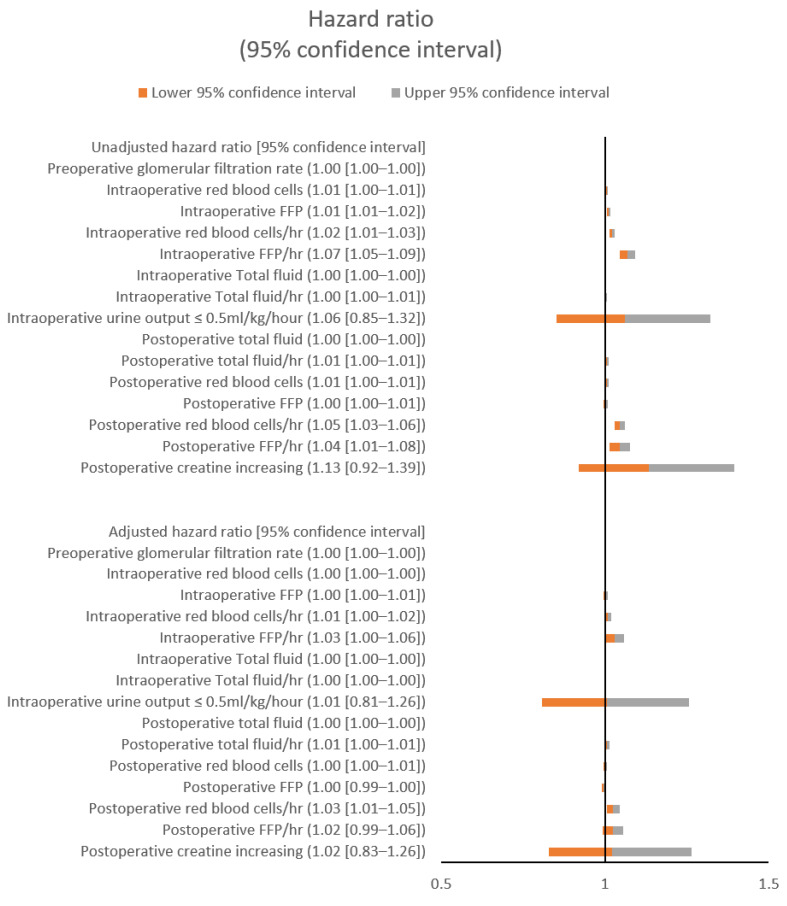
Unadjusted and adjusted hazard ratios for postoperative pulmonary edema. FFP: fresh frozen plasma.

**Figure 2 jcm-10-04224-f002:**
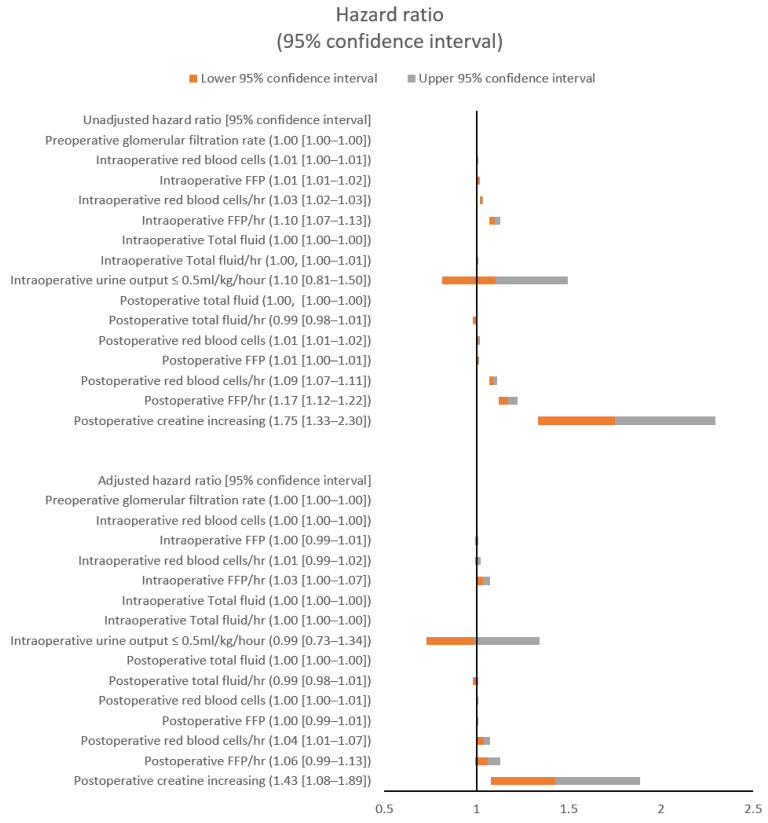
Unadjusted and adjusted hazard ratios for postoperative pulmonary edema with hypoxemia. FFP: fresh frozen plasma.

**Table 1 jcm-10-04224-t001:** Demographic and clinical data of postoperative pulmonary edema patients.

Variable		No Postoperative Pulmonary Edema (*n* = 1790)	Postoperative Pulmonary Edema (*n* = 300)	*p*-Value
**Demographics**				
	Old age, *n* (%)	496 (27.7)	108 (36.0)	0.003
	Males, *n* (%)	904 (50.5)	148 (49.3)	0.71
	Obesity, *n* (%)	68 (3.8)	11 (3.7)	0.91
**Preoperative clinical features**				
	Emergency, *n* (%)	647 (36.1)	148 (49.3)	<0.001
	American Society of Anesthesiologists physical status > 2, *n* (%)	862 (48.2)	186 (62.0)	<0.001
	Tobacco use, *n* (%)	415 (23.2)	62 (20.7)	0.346
	Brain trauma, *n* (%)	230 (12.8)	42 (14.0)	0.58
	Multiple fractures, *n* (%)	127 (7.1)	33 (11.0)	0.02
	Hyponatremia, *n* (%)	240 (13.4)	34 (11.3)	0.33
	Hypoalbuminemia, *n* (%)	542 (30.3)	117 (39.0)	0.003
	Glomerular filtration rate, median (interquartile range), mL/min/1.73 m^2^	93.4 (72.7–93.4)	84.4 (68.3–84.4)	<0.001
**Intraoperative clinical features**				
	General anesthesia, *n* (%)	1730 (96.6)	299 (99.7)	0.004
	Anesthesia time, median (interquartile range), h	5.0 (3.5–5.0)	5.1 (2.9–5.1)	0.54
	Acute abdomen surgery, *n* (%)	119 (6.6)	34 (11.3)	0.004
	Aorta surgery, *n* (%)	30 (1.7)	7 (2.3)	0.42
	Brain surgery, *n* (%)	391 (21.8)	59 (19.7)	0.40
	Spine surgery, *n* (%)	270 (15.1)	58 (19.3)	0.06
	Thoracic surgery, *n* (%)	53 (3.0)	13 (4.3)	0.21
	Massive transfusion, *n* (%)	57 (3.2)	33 (11.0)	<0.001
	Urine output ≤ 0.5 mL/kg/h, *n* (%)	202 (11.3)	39 (13.0)	0.39
	Continuous inotrope use, *n* (%)	375 (20.9)	96 (32.0)	<0.001
	Red blood cells, median (interquartile range), %	24.3 (12.1–24.3)	32.9 (20.9–32.9)	<0.001
	Red blood cells per hour, median (interquartile range), %/h	4.6 (2.0–4.6)	6.3 (3.7–6.3)	<0.001
	FFP, median (interquartile range), %	1.5 (0.0–1.5)	9.5 (0.0–9.5)	<0.001
	FFP per hour, median (interquartile range), %/h	0.1 (0.0–0.1)	1.7 (0.0–1.7)	<0.001
	Total fluid, median (interquartile range), %	113.7 (87.9–113.7)	119.7 (89.7–119.7)	0.04
	Total fluid per hour, median (interquartile range), %/h	22.6 (17.2–22.6)	24.3 (17.8–24.3)	0.01
	Estimated blood loss, median (interquartile range), %	54.4 (45.5–54.4)	63.0 (48.6–63.0)	<0.001
**Postoperative clinical features**				
	Patient controlled analgesia, *n* (%)	1139 (63.6)	181 (60.3)	0.27
**Time-varying postoperative clinical features**				
**Period 1** (No postoperative pulmonary edema, *n* = 1866; postoperative pulmonary edema, *n* = 192)	Start time, median (interquartile range), h	0.0 (0.0–0.0)	0.0 (0.0–0.0)	1.000
	End time, median (interquartile range), h	10.4 (5.6–16.1)	0.8 (0.3–6.2)	<0.001
	Total fluid, median (interquartile range), %	29.0 (10.2–58.2)	2.4 (0.0–24.5)	<0.001
	Total fluid per hour, median (interquartile range), %/h	2.8 (1.3–4.7)	1.5 (0.0–4.4)	<0.001
	Red blood cells, median (interquartile range), %	0.0 (0.0–0.0)	0.0 (0.0–0.0)	0.79
	FFP, median (interquartile range), %	0.0 (0.0–0.0)	0.0 (0.0–0.0)	0.76
	Red blood cells per hour, median (interquartile range), %/h	0.0 (0.0–0.0)	0.0 (0.0–0.0)	0.34
	FFP per hour, median (interquartile range), %/h	0.0 (0.0–0.0)	0.0 (0.0–0.0)	0.88
	Creatinine increase, *n* (%)	190 (10.2)	20 (10.4)	0.92
**Period 2** (No postoperative pulmonary edema, *n* = 20; postoperative pulmonary edema, *n* = 1)	Start time, median (interquartile range), h	0.0 (0.0–0.0)	0.0 (0.0–0.0)	1.000
	End time, median (interquartile range), h	33.8 (29.2–37.2)	43.4 (43.4–43.4)	0.48
	Total fluid, median (interquartile range), %	69.9 (42.7–119.4)	25.6 (25.6–25.6)	0.38
	Total fluid per hour, median (interquartile range), %/h	2.5 (1.5–3.4)	0.6 (0.6–0.6)	0.29
	Red blood cells, median (interquartile range), %	0.0 (0.0–12.8)	5.0 (5.0–5.0)	0.86
	FFP, median (interquartile range), %	0.0 (0.0–4.5)	0.0 (0.0–0.0)	0.67
	Red blood cells per hour, median (interquartile range), %/h	0.0 (0.0–0.4)	0.1 (0.1–0.1)	0.86
	FFP per hour, median (interquartile range), %/h	0.0 (0.0–0.1)	0.0 (0.0–0.0)	0.67
	Creatinine increase, *n* (%)	2 (10.0)	1 (100.0)	0.14
**Period 3** (No postoperative pulmonary edema, *n* = 11; postoperative pulmonary edema, *n* = 0)	Start time, median (interquartile range), h	0.0 (0.0–0.0)		
	End time, median (interquartile range), h	59.2 (58.3–64.1)		
	Total fluid, median (interquartile range), %	74.8 (22.0–140.3)		
	Total fluid per hour, median (interquartile range), %/h	1.2 (0.4–2.3)		
	Red blood cells, median (interquartile range), %	0.0 (0.0–0.0)		
	FFP, median (interquartile range), %	0.0 (0.0–0.0)		
	Red blood cells per hour, median (interquartile range), %/h	0.0 (0.0–0.0)		
	FFP per hour, median (interquartile range), %/h	0.0 (0.0–0.0)		
	Creatinine increase, *n* (%)	2 (18.2)		
**Period 4** (No postoperative pulmonary edema, *n* = 1301; postoperative pulmonary edema, *n* = 63)	Start time, median (interquartile range), h	11.4 (7.0–17.1)	11.0 (8.8–16.0)	0.73
	Start time, median (interquartile range), h	35.6 (31.1–41.2)	34.2 (30.9–37.5)	0.08
	End time, median (interquartile range), h	106.1 (68.3–163.5)	111.1 (77.9–158.9)	0.29
	Total fluid, median (interquartile range), %	4.6 (2.8–7.2)	5.0 (3.7–7.7)	0.09
	Total fluid per hour, median (interquartile range), %/h	0.0 (0.0–10.3)	9.5 (0.0–25.9)	<0.001
	Red blood cells, median (interquartile range), %	0.0 (0.0–7.0)	4.0 (0.0–17.9)	<0.001
	FFP, median (interquartile range), %	0.0 (0.0–0.4)	0.4 (0.0–1.5)	<0.001
	Red blood cells per hour, median (interquartile range), %/h	0.0 (0.0–0.3)	0.2 (0.0–0.8)	<0.001
	FFP per hour, median (interquartile range), %/h	198 (15.2)	16 (25.4)	0.03
**Period 5** (No postoperative pulmonary edema, *n* = 81; postoperative pulmonary edema, *n* = 2)	Start time, median (interquartile range), h	10.3 (5.1–15.7)	9.6 (9.2–10.0)	0.78
	End time, median (interquartile range), h	58.5 (53.8–64.6)	55.9 (54.8–57.0)	0.49
	Total fluid, median (interquartile range), %	89.2 (60.9–160.2)	124.3 (111.2–137.5)	0.53
	Total fluid per hour, median (interquartile range), %/h	2.0 (1.4–3.2)	2.7 (2.4–2.9)	0.43
	Red blood cells, median (interquartile range), %	0.0 (0.0–10.7)	2.6 (1.3–4.0)	0.81
	FFP, median (interquartile range), %	0.0 (0.0–0.0)	1.8 (0.9–2.7)	0.53
	Red blood cells per hour, median (interquartile range), %/h	0.0 (0.0–0.2)	0.1 (0.0–0.1)	0.81
	FFP per hour, median (interquartile range), %/h	0.0 (0.0–0.0)	0.0 (0.0–0.1)	0.53
	Creatinine increase, *n* (%)	1 (1.2)	0 (0.0)	>0.999
**Period 6** (No postoperative pulmonary edema, *n* = 1008; postoperative pulmonary edema, *n* = 42)	Start time, median (interquartile range), h	35.3 (30.9–40.9)	35.4 (30.9–41.5)	0.96
	End time, median (interquartile range), h	59.0 (54.9–64.9)	58.2 (52.4–63.1)	0.21
	Total fluid, median (interquartile range), %	178.7 (109.1–268.1)	166.5 (108.0–237.0)	0.46
	Total fluid per hour, median (interquartile range), %/h	7.7 (4.6–11.5)	6.9 (4.9–11.1)	0.79
	Red blood cells, median (interquartile range), %	0.0 (0.0–13.9)	11.6 (0.0–25.2)	0.001
	FFP, median (interquartile range), %	0.0 (0.0–8.3)	0.0 (0.0–14.4)	0.19
	Red blood cells per hour, median (interquartile range), %/h	0.0 (0.0–0.6)	0.4 (0.0–1.2)	0.001
	FFP per hour, median (interquartile range), %/h	0.0 (0.0–0.4)	0.0 (0.0–0.6)	0.17
	Creatinine increase, *n* (%)	153 (15.2)	13 (31.0)	0.01

Period 1: from end of anesthesia to 24 h after anesthesia; period 2: from end of anesthesia to 48 h post anesthesia; period 3: from end of anesthesia to 72 h after anesthesia; period 4: 24–48 h after anesthesia; period 5: 24–72 h after anesthesia; period 6: 48–72 h after anesthesia. Red blood cells, FFP, total fluid, and estimated blood loss are expressed as ratios relative to the average blood volume. FFP: fresh frozen plasma.

**Table 2 jcm-10-04224-t002:** Demographic and clinical data of postoperative pulmonary edema patients with hypoxemia.

Variable		No Postoperative Pulmonary Edema with Hypoxemia (*n* = 1499)	Postoperative Pulmonary Edema with Hypoxemia (*n* = 161)	*p*-Value
**Demographics**				
	Old age, *n* (%)	384 (25.6)	63 (39.1)	<0.001
	Males, *n* (%)	730 (48.7)	88 (54.7)	0.15
	Obesity, *n* (%)	46 (3.1)	6 (3.7)	0.65
**Preoperative clinical features**				
	Emergency, *n* (%)	516 (34.4)	89 (55.3)	<0.001
	American Society of Anesthesiologists physical status > 2, *n* (%)	698 (46.6)	117 (72.7)	<0.001
	Tobacco use, *n* (%)	332 (22.1)	36 (22.4)	0.95
	Brain trauma, *n* (%)	183 (12.2)	24 (14.9)	0.33
	Multiple fracture, *n* (%)	106 (7.1)	19 (11.8)	0.03
	Hyponatremia, *n* (%)	195 (13.0)	21 (13.0)	0.99
	Hypoalbuminemia, *n* (%)	443 (29.6)	69 (42.9)	0.001
	Glomerular filtration rate, median (interquartile range), mL/min/1.73 m^2^	94.1 (72.8–122.9)	83.7 (65.4–108.6)	<0.001
**Intraoperative clinical features**				
	General anesthesia, *n* (%)	1445 (96.4)	160 (99.4)	0.05
	Anesthesia time, median (interquartile range), h	5.0 (3.5–6.9)	4.7 (2.8–7.2)	0.58
	Acute abdomen surgery, *n* (%)	97 (6.5)	26 (16.1)	<0.001
	Aorta surgery, *n* (%)	12 (0.8)	5 (3.1)	0.01
	Brain surgery, *n* (%)	319 (21.3)	35 (21.7)	0.89
	Spine surgery, *n* (%)	235 (15.7)	25 (15.5)	0.96
	Thoracic surgery, *n* (%)	41 (2.7)	6 (3.7)	0.47
	Massive transfusion, *n* (%)	35 (2.3)	25 (15.5)	<0.001
	Urine output ≤ 0.5 mL/kg/h, *n* (%)	154 (10.3)	21 (13.0)	0.28
	Continuous inotropes use, *n* (%)	279 (18.6)	62 (38.5)	<0.001
	Red blood cells, median (interquartile range), %	23.7 (11.8–37.2)	34.6 (21.1–61.0)	<0.001
	Red blood cells per hour, median (interquartile range), %/h	4.6 (1.9–8.0)	6.9 (3.8–16.7)	<0.001
	FFP, median (interquartile range), %	0.0 (0.0–10.1)	9.7 (0.0–18.6)	<0.001
	FFP per hour, median (interquartile range), %/h	0.0 (0.0–2.1)	1.9 (0.0–5.4)	<0.001
	Total fluid, median (interquartile range), %	113.6 (87.8–146.9)	126.9 (91.9–178.0)	0.02
	Total fluid per hour, median (interquartile range), %	22.8 (17.3–31.6)	26.5 (18.0–40.1)	0.02
	Estimated blood loss, median (interquartile range), %	54.3 (45.5–72.6)	66.7 (51.0–104.2)	<0.001
**Postoperative clinical features**				
	Patient controlled analgesia, *n* (%)	979 (65.3)	83 (51.6)	0.001
**Time-varying postoperative clinical features**				
**Period 1** (No postoperative pulmonary edema with hypoxemia, *n* = 1275; postoperative pulmonary edema with hypoxemia, *n* = 72)	Start time, median (interquartile range), h	0.0 (0.0–0.0)	0.0 (0.0–0.0)	>0.999
	End time, median (interquartile range), h	9.9 (3.3–15.3)	1.3 (0.8–7.1)	<0.001
	Total fluid, median (interquartile range), %	27.1 (5.4–54.5)	0.9 (0.0–22.4)	<0.001
	Total fluid per hour, median (interquartile range), %/h	2.7 (1.1–4.5)	0.3 (0.0–2.8)	<0.001
	Red blood cells, median (interquartile range), %	0.0 (0.0–0.0)	0.0 (0.0–7.1)	0.001
	FFP, median (interquartile range), %	0.0 (0.0–0.0)	0.0 (0.0–5.3)	0.004
	Red blood cells per hour, median (interquartile range), %/h	0.0 (0.0–0.0)	0.0 (0.0–1.7)	<0.001
	FFP per hour, median (interquartile range), %/h	0.0 (0.0–0.0)	0.0 (0.0–0.6)	0.002
	Creatinine increase, *n* (%)	99 (7.8)	11 (15.3)	0.02
**Period 2** (No postoperative pulmonary edema with hypoxemia, *n* = 148; postoperative pulmonary edema with hypoxemia, *n* = 42)	Start time, median (interquartile range), h	0.0 (0.0–0.0)	0.0 (0.0–0.0)	>0.999
	End time, median (interquartile range), h	37.2 (31.3–41.6)	35.3 (30.0–39.6)	0.12
	Total fluid, median (interquartile range), %	102.3 (70.7–150.7)	102.9 (54.0–145.5)	0.76
	Total fluid per hour, median (interquartile range), %/h	2.9 (1.9–4.1)	2.8 (1.6–4.4)	0.93
	Red blood cells, median (interquartile range), %	0.0 (0.0–7.4)	0.0 (0.0–21.2)	0.05
	FFP, median (interquartile range), %	0.0 (0.0–0.0)	7.9 (0.0–21.6)	<0.001
	Red blood cells per hour, median (interquartile range), %/h	0.0 (0.0–0.2)	0.0 (0.0–0.6)	0.04
	FFP per hour, median (interquartile range), %/h	0.0 (0.0–0.0)	0.3 (0.0–0.6)	<0.001
	Creatinine increase, *n* (%)	17 (11.5)	19 (45.2)	<0.001
**Period 3** (No postoperative pulmonary edema with hypoxemia, *n* = 86; postoperative pulmonary edema with hypoxemia *n* = 21)	Start time, median (interquartile range), h	0.0 (0.0–0.0)	0.0 (0.0–0.0)	>0.999
	End time, median (interquartile range), h	59.1 (55.3–64.8)	57.2 (54.5–61.2)	0.20
	Total fluid, median (interquartile range), %	132.7 (75.9–222.2)	198.8 (118.1–279.2)	0.05
	Total fluid per hour, median (interquartile range), %/h	4.3 (0.0–21.7)	38.5 (10.0–82.3)	0.04
	Red blood cells, median (interquartile range), %	2.2 (1.3–3.8)	3.4 (2.1–5.0)	0.01
	FFP, median (interquartile range), %	0.1 (0.0–0.3)	0.6 (0.2–1.4)	0.01
	Red blood cells per hour, median (interquartile range), %/h	0.0 (0.0–14.0)	11.0 (0.0–44.0)	0.01
	FFP per hour, median (interquartile range), %/h	0.0 (0.0–8.5)	8.3 (0.0–40.0)	0.01
	Creatinine increase, *n* (%)	9 (10.5)	7 (33.3)	0.01
**Period 4** (No postoperative pulmonary edema with hypoxemia, *n* = 583; postoperative pulmonary edema with hypoxemia, *n* = 0)	Start time, median (interquartile range), h	11.4 (5.6–16.0)		
	End time, median (interquartile range), h	36.9 (31.9–40.8)		
	Total fluid, median (interquartile range), %	116.0 (78.3–178.2)		
	Total fluid per hour, median (interquartile range), %/h	0.0 (0.0–20.7)		
	Red blood cells, median (interquartile range), %	4.6 (3.0–7.4)		
	FFP, median (interquartile range), %	0.0 (0.0–0.8)		
	Red blood cells per hour, median (interquartile range), %/h	0.0 (0.0–10.2)		
	FFP per hour, median (interquartile range), %/h	0.0 (0.0–7.9)		
	Creatinine increase, *n* (%)	71 (12.2)		
**Period 5** (No postoperative pulmonary edema with hypoxemia, *n* = 148; postoperative pulmonary edema with hypoxemia, *n* = 12)	Start time, median (interquartile range), h	10.5 (5.5–15.5)	9.4 (7.1–16.9)	0.66
	End time, median (interquartile range), h	60.0 (55.6–64.3)	59.5 (55.1–63.2)	0.77
	Total fluid, median (interquartile range), %	137.2 (79.0–211.5)	227.4 (186.0–284.8)	0.003
	Total fluid per hour, median (interquartile range), %/h	0.0 (0.0–16.9)	22.3 (0.0–41.4)	0.003
	Red blood cells, median (interquartile range), %	2.8 (1.7–4.4)	5.0 (3.5–5.9)	0.39
	FFP, median (interquartile range), %	0.0 (0.0–0.4)	0.5 (0.0–0.9)	0.03
	Red blood cells per hour, median (interquartile range), %/h	0.0 (0.0–13.9)	5.9 (0.0–14.0)	0.32
	FFP per hour, median (interquartile range), %/h	0.0 (0.0–5.2)	6.8 (0.0–24.2)	0.03
	Creatinine increase, *n* (%)	14 (9.5)	5 (41.7)	0.001
**Period 6** (No postoperative pulmonary edema with hypoxemia, *n* = 384; postoperative pulmonary edema with hypoxemia, *n* = 14)	Start time, median (interquartile range), h	37.3 (32.2–41.3)	36.6 (32.9–38.6)	0.63
	End time, median (interquartile range), h	60.9 (55.6–65.6)	60.1 (57.3–63.2)	0.61
	Total fluid, median (interquartile range), %	204.7 (125.8–297.4)	172.5 (135.9–218.7)	0.58
	Total fluid per hour, median (interquartile range), %/h	7.5 (0.0–29.0)	30.1 (15.6–68.9)	0.56
	Red blood cells, median (interquartile range), %	8.6 (5.3–12.8)	7.1 (5.5–9.0)	>0.999
	FFP, median (interquartile range), %	0.3 (0.0–1.3)	1.4 (0.5–2.8)	0.27
	Red blood cells per hour, median (interquartile range), %/h	0.0 (0.0–14.0)	18.4 (14.7–37.0)	>0.999
	FFP per hour, median (interquartile range), %/h	0.0 (0.0–13.3)	2.6 (0.0–22.9)	0.30
	Creatinine increase, *n* (%)	47 (12.2)	4 (28.6)	0.07

Period 1: from end of anesthesia to 24 h after anesthesia; period 2: from end of anesthesia to 48 h post anesthesia; period 3: from end of anesthesia to 72 h after anesthesia; period 4: 24–48 h after anesthesia; period 5: 24–72 h after anesthesia; period 6: 48–72 h after anesthesia. Red blood cells, FFP, total fluid and estimated blood loss are expressed as ratio for average blood volume; FFP: fresh frozen plasma.

**Table 3 jcm-10-04224-t003:** Blood and fluid cut-off values for the development of pulmonary edema and pulmonary edema with hypoxemia.

Pulmonary Edema	Cut-Off Value	ROC-AUC	95% CI	*p*-Value
**Intraop. RBCs**	27.7	0.61	0.58–0.64	<0.001
**Intraop. FFP**	9.0	0.62	0.59–0.65	<0.001
**Intraop. RBCs/h**	3.7	0.61	0.58–0.64	<0.001
**Intraop. FFP/h**	2.4	0.62	0.59–0.65	<0.001
**Intraop. total fluid**	120.7	0.53	0.50–0.56	0.07
**Intraop. total fluid/h**	21.5	0.55	0.52–0.58	<0.001
**Postop. total fluid**	579.0	0.35	0.32–0.38	<0.001
**Postop. total fluid/h**	11.8	0.44	0.40–0.47	<0.001
**Postop. RBCs**	15.2	0.53	0.50–0.56	0.02
**Postop. FFP**	9.1	0.52	0.49–0.55	0.17
**Postop. RBCs/h**	1.2	0.54	0.51–0.58	<0.001
**Postop. FFP/h**	0.5	0.53	0.50–0.56	0.06
**Pulmonary edema with hypoxemia**				
**Intraop. RBCs**	239.6	0.65	0.61–0.70	<0.001
**Intraop. FFP**	87.4	0.64	0.59–0.68	<0.001
**Intraop. RBCs/h**	54.5	0.65	0.61–0.70	<0.001
**Intraop. FFP/h**	15.1	0.64	0.59–0.69	<0.001
**Intraop. total fluid**	510.8	0.56	0.51–0.61	0.01
**Intraop. total fluid/h**	124.3	0.57	0.52–0.62	<0.001
**Postop. total fluid**	718.9	0.48	0.43–0.53	0.36
**Postop. total fluid/h**	10.0	0.41	0.36–0.45	<0.001
**Postop. RBCs**	100.5	0.61	0.56–0.66	<0.001
**Postop. FFP**	123.3	0.61	0.56–0.66	<0.001
**Postop. RBCs/h**	13.3	0.61	0.56–0.66	<0.001
**Postop. FFP/h**	8.4	0.60	0.55–0.65	<0.001

Intraop., intraoperative; postop., postoperative; FFP, fresh frozen plasma; RBCs, red blood cells; CI, confidence interval; ROC, receiver operating characteristic; AUC, area under curve.

**Table 4 jcm-10-04224-t004:** Adjusted hazard ratios for postoperative pulmonary edema and pulmonary edema with hypoxemia during emergency surgery.

Pulmonary Edema	HR	95% CI	*p*-Value
**Intraop. RBCs**	1.00	1.00–1.00	0.96
**Intraop. FFP**	1.00	0.99–1.01	0.91
**Intraop. RBCs/h**	1.01	1.00–1.02	0.24
**Intraop. FFP/h**	1.03	1.00–1.07	0.05
**Intraop. total fluid**	1.00	1.00–1.00	0.65
**Intraop. total fluid/h**	1.00	1.00–1.00	0.63
**Oliguria**	1.05	0.83–1.33	0.66
**Preop. GFR**	1.00	1.00–1.00	0.38
**Postop. total fluid**	1.00	1.00–1.00	0.00
**Postop. total fluid/h**	1.00	0.99–1.01	0.65
**Postop. RBCs**	1.00	0.99–1.01	0.97
**Postop. FFP**	1.00	0.99–1.00	0.32
**Postop. RBCs/h**	1.02	1.00–1.05	0.05
**Postop. FFP/h**	1.02	0.99–1.06	0.16
**Postop. Cr increase**	1.03	0.77–1.38	0.83
**Pulmonary edema with hypoxemia**			
**Intraop. RBCs**	1.00	0.99–1.00	0.85
**Intraop. FFP**	1.00	0.99–1.01	0.87
**Intraop. RBCs/h**	1.00	0.99–1.02	0.70
**Intraop. FFP/h**	1.03	0.99–1.08	0.14
**Intraop. total fluid**	1.00	1.00–1.00	0.78
**Intraop. total fluid/h**	1.00	1.00–1.01	0.76
**Oliguria**	1.35	0.80–2.27	0.26
**Preop. GFR**	1.00	0.99–1.00	0.11
**Postop. total fluid**	1.00	1.00–1.00	0.00
**Postop. total fluid/h**	1.00	0.97–1.02	0.80
**Postop. RBCs**	1.00	1.00–1.01	0.36
**Postop. FFP**	1.00	0.99–1.01	0.81
**Postop. RBCs/h**	1.03	0.99–1.07	0.11
**Postop. FFP/h**	1.06	0.99–1.14	0.10
**Postop. Cr increase**	1.39	0.94–2.05	0.10

Intraop., intraoperative; preop., preoperative; postop., postoperative; FFP, fresh frozen plasma; RBCs, red blood cell; Cr, creatinine; CI, confidence interval; GFR, glomerular filtration rate. HR: hazard ratio

## Data Availability

The datasets used and/or analyzed during the current study are available from the corresponding author upon reasonable request.

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
