# Peer review of "Effect of Intra- and Post-Operative Fluid and Blood Volume on Postoperative Pulmonary Edema in Patients with Intraoperative Massive Bleeding"

_jcm, 2021, doi:10.3390/jcm10184224_

Round 1

Reviewer 1 Report

Overall a lot of work and effort from the authors.  The paper attempts to elucidate the risks of resuscitation with regards to post operative pulmonary edema.  While the premise of the article is sound, I have a few major issues with the basis of the manuscript.

1). What was the resuscitation strategy of the anesthesiologists intra-operatively?  When the patient was bleeding, was their strategy a 1:1:1 resuscitation strategy (blood, FFP and platelets in that ratio?)?  Platelets are absolutely necessary for coagulation and the authors have no mention of platelet administration.  Some say the earlier the platelets the better.  If the premise of the article is to determine if massively bleeding, resuscitated patients had post operative pulmonary edema - you must describe how the patient was resuscitated.  Did the providers go by base excess, urine output, blood pressure?  What were the blood pressure drops in the operating room - as this can help determine multi system failure.  There are a lot of questions I still have with regards to how the resuscitation was performed.

2). I fail to understand how brain surgery, aortic surgery and emergency surgery can all be lumped together.  These are all very different types of surgery with different repercussions.  If you have a massively bleeding patient from brain surgery - I would wonder what happened in the operating room versus aortic surgery.  I do not think it is appropriate to have all these types of operations together.  

3). The authors allude to this in the limitations with regards to not knowing the patient's pre operative cardiac status.  This to me is a massive limitation making the paper hard to interpret.  In some cases, a patient make have baseline pulmonary edema from mild heart failure but who needs aortic surgery.  This is incredibly important to know if you are using CXR as a determinant for pulmonary edema.

There were a handful of other issues with the paper - but these were the main ones that if corrected appropriately, may make the paper helpful in the literature.

Author Response

Overall a lot of work and effort from the authors.  The paper attempts to elucidate the risks of resuscitation with regards to post operative pulmonary edema.  While the premise of the article is sound, I have a few major issues with the basis of the manuscript.

1). What was the resuscitation strategy of the anesthesiologists intra-operatively?  When the patient was bleeding, was their strategy a 1:1:1 resuscitation strategy (blood, FFP and platelets in that ratio?)?  Platelets are absolutely necessary for coagulation and the authors have no mention of platelet administration.  Some say the earlier the platelets the better.  If the premise of the article is to determine if massively bleeding, resuscitated patients had post operative pulmonary edema - you must describe how the patient was resuscitated.  Did the providers go by base excess, urine output, blood pressure?  What were the blood pressure drops in the operating room - as this can help determine multi system failure.  There are a lot of questions I still have with regards to how the resuscitation was performed.

 Answer

Thank you for your comment. Our study’s subject is patients with intraoperative massive bleeding (>2000ml), not patients with massive transfusion. Patients with bleeding of 2000ml always don’t need massive transfusion. Approximately 25% of patients in our study had less than 2100ml intraoperative bleeding. If oxygen supply is enough and vital sign is stable, massive transfusion is not required. In case of bleeding, crystalloid and colloid are administered first. The amount of transfusion is not equal to the amount of blood loss. Transfusion is performed when applicable to transfusion indication.

Also, thank you for advising for platelet. We think that 1:1:1 method is best in massive transfusion and perform 1:1:1 method in massive transfusion. However, when massive transfusion is not required as above, we did not always perform platelet transfusion. So, we did not include platelets in our data.

Thank you for good comments for urine output, blood pressure and base excess.

In urine output, there were many cases not to measure urine output per hour due to no urine catheter, especially when recovered patients did self-voiding in the postoperative period.  Measuring interval of blood pressure was too variable according to the condition of patients and because it was difficult to match measuring time with AGBA and chest x-ray, we could not include it as variable. Also, because intraoperative blood pressure is recorded by image, we could not extract blood pressure in big data. We hadn't thought of base excess. As you're advising, if we include platelet, urine output, blood pressure and base excess, we think that the content of our article may be much better. However, it takes time to apply to get new data, and we cannot get data from 2009 to 2010 because the preservation period of medical records is 10 years in Korea. If we conduct follow-up study, we will apply your advice. Output, blood pressure, and base excess are described in limitation.

2). I fail to understand how brain surgery, aortic surgery and emergency surgery can all be lumped together.  These are all very different types of surgery with different repercussions.  If you have a massively bleeding patient from brain surgery - I would wonder what happened in the operating room versus aortic surgery.  I do not think it is appropriate to have all these types of operations together.

Answer:

Thanks for the good advice. We wanted to investigate the effect of fluids and blood on pulmonary edema during and after surgery in general, but you are right. However, some surgeries do not have a sufficient number of samples for analysis. There are not enough cases of aortic and brain surgery for the number of covariates included in the analysis. 10% of the number of above two surgical cases is less than the number of covariates. In particular, the number of cases of aortic surgery was so small that only 37 cases were performed. However, for emergency surgery, an analysis was added, and the contents were also added.

3). The authors allude to this in the limitations with regards to not knowing the patient's pre operative cardiac status.  This to me is a massive limitation making the paper hard to interpret.  In some cases, a patient make have baseline pulmonary edema from mild heart failure but who needs aortic surgery.  This is incredibly important to know if you are using CXR as a determinant for pulmonary edema.

Answer:

Thanks for the good comments. A common cause of pulmonary edema is that of the heart. Not knowing the condition of the patient's heart before surgery means that we may not know if there is a causative factor. If we knew the patient's preoperative heart condition, we would have included it in the study as a risk factor of pulmonary edema. The purpose of this study was to investigate the risk of fluids and transfusions after surgery and postoperatively. Including patients with preoperative pulmonary edema, even with mild pulmonary edema, seems to be different from the purpose of our study because it should be considered as a worsening factor rather than a risk factor. As you say, diagnosing pulmonary edema with a chest x-ray can be inaccurate. So, hypoxemia was considered and analyzed to evaluate the objectivity and severity of pulmonary edema symptoms. We have added it to a limitation.

Reviewer 2 Report

Manuscript ID: jcm-1351552

Kwon et al.

In this manuscript the authors study the effects of intra- and postoperative fluid and blood product replacements on the development of pulmonary edema (with and without hypoxemia) in patients with massive intraoperative bleeding (>40% of blood volume). Not too surprisingly, patients of older age, higher ASA physical status, received larger volumes of intra- and postoperative fluids and blood products, and had worse kidney function had a higher risk of developing pulmonary edema, and the majority of these associations remained in place for the risk of pulmonary edema with hypoxemia.

Main strengths of the study consist in the relatively large study population (2090 patients over a 10-year period), and the distinction of clinically insignificant pulmonary edema and pulmonary edema with hypoxemia requiring intervention.

One of the main weaknesses of the study is the fairly predictable outcome. I would also encourage the authors to attempt providing a cut-off value for fluid and blood product volumes above which clinically significant (severe) pulmonary edema occurs. Such a group could for example consist of patients requiring more than simple oxygen supplementation (i.e. mechanically ventilated or high flow nasal cannula/Bipap/CPAP). I would also encourage the authors to define whether or not the patients with elevated creatinine levels and decreased urine output (positive association with pulmonary edema development) actually met Acute Kidney Injury (AKI) criteria. Could the authors also mention if patient data were de-identified?

Additionally, the manuscript would benefit from proofreading since it contains multiple duplications in wording and misplaced reference numbers.

Author Response

In this manuscript the authors study the effects of intra- and postoperative fluid and blood product replacements on the development of pulmonary edema (with and without hypoxemia) in patients with massive intraoperative bleeding (>40% of blood volume). Not too surprisingly, patients of older age, higher ASA physical status, received larger volumes of intra- and postoperative fluids and blood products, and had worse kidney function had a higher risk of developing pulmonary edema, and the majority of these associations remained in place for the risk of pulmonary edema with hypoxemia.

Main strengths of the study consist in the relatively large study population (2090 patients over a 10-year period), and the distinction of clinically insignificant pulmonary edema and pulmonary edema with hypoxemia requiring intervention.

One of the main weaknesses of the study is the fairly predictable outcome. I would also encourage the authors to attempt providing a cut-off value for fluid and blood product volumes above which clinically significant (severe) pulmonary edema occurs. Such a group could for example consist of patients requiring more than simple oxygen supplementation (i.e. mechanically ventilated or high flow nasal cannula/Bipap/CPAP). I would also encourage the authors to define whether or not the patients with elevated creatinine levels and decreased urine output (positive association with pulmonary edema development) actually met Acute Kidney Injury (AKI) criteria. Could the authors also mention if patient data were de-identified?

Answer:

Thanks for the good comments.

  1. We added a cutoff value.
  2. Diagnosis of AKI is follow:

An abrupt (within 48 hours) reduction in kidney function currently defined as an absolute increase in serum creatinine of more than or equal to 0.3 mg/dl (≥ 26.4 μmol/l), a percentage increase in serum creatinine of more than or equal to 50% (1.5-fold from baseline), or a reduction in urine output (documented oliguria of less than 0.5 ml/kg per hour for more than six hours).

Although creatine was used in our study, the definition of AKI is a little different. The time that applies to AKI is defined as within 48 hours, but in our cases, it is not an exact match as we include patients over 48 hours because including patients measured creatinine in postoperative 48-72 hour. This is described in the manuscript.

  1. When requesting data, all data is provided de-identified. We can extract various data of the same person through the key. This was mentioned in the methods of text.

Additionally, the manuscript would benefit from proofreading since it contains multiple duplications in wording and misplaced reference numbers.

Answer:

English proofreading of manuscript was done.

Round 2

Reviewer 2 Report

I applaud the authors for their substantial revisions. My only remaining concern that somewhat decreases my enthusiasm remains the fairly predictable outcome of the study.